# Safety and immunogenicity of a phase 1/2 randomized clinical trial of a quadrivalent, mRNA-based seasonal influenza vaccine (mRNA-1010) in healthy adults: interim analysis

Ivan T. Lee[1], Raffael Nachbagauer [ORCID][1] ✉, David Ensz[2], Howard Schwartz[3], Lizbeth Carmona[1], Kristi Schaefers[1], Andrei Avanesov[1], Daniel Stadlbauer[1], Carole Henry[1], Ren Chen[1], Wenmei Huang[1], Daniela Ramirez Schrempp[1], Jintanat Ananworanich[1] & Robert Paris[1] ✉

Despite vaccine availability, influenza remains a substantial global public health concern. Here, we report interim findings on the primary and secondary objectives of the safety, reactogenicity, and humoral immunogenicity of a quadrivalent messenger RNA (mRNA) vaccine against seasonal influenza, mRNA-1010, from the first 2 parts of a 3-part, first-in-human, phase 1/2 clinical trial in healthy adults aged ≥18 years (NCT04956575). In the placebo-controlled Part 1, a single dose of mRNA-1010 (50 μg, 100 μg, or 200 μg) elicited hemagglutination inhibition (HAI) titers against vaccine-matched strains. In the active-comparator-controlled Part 2, mRNA-1010 (25 μg, 50 μg, or 100 μg) elicited higher HAI titers than a standard dose, inactivated seasonal influenza vaccine for influenza A strains and comparable HAI titers for influenza B strains. No safety concerns were identified; solicited adverse reactions were dose-dependent and more frequent after receipt of mRNA-1010 than the active comparator. These interim data support continued development of mRNA-1010.

Vaccination remains an essential public health strategy for the prevention of influenza[1]. However, the effectiveness of currently available influenza vaccines varies each season and is impacted by mismatch with circulating influenza strains, poor vaccine uptake, and weak, short-lived immunity[2–4]. Consequently, the burden of disease remains substantial worldwide[2,5]. Seasonal influenza epidemics occur annually, following a temporal circulation pattern, with increased cases during the winter months[6]. According to the World Health Organization (WHO), seasonal influenza is estimated to cause 3 to 5 million cases of severe illness and up to 650,000 deaths worldwide each year[2]. While influenza affects all age groups, adults with chronic medical conditions[7] and older adults[8] are at increased risk of serious influenza-associated complications, with 67% of influenza deaths occurring in those ≥65 years of age[9].

Four types of influenza viruses exist: A, B, C, and D; however, only A and B are responsible for the majority of human illness, with influenza A the major cause of severe disease and death in older adults[2,9–11]. The main surface glycoprotein of influenza viruses is hemagglutinin

[1]Moderna, Inc., Cambridge, MA, USA. [2]Meridian Clinical Research, Sioux City, IA, USA. [3]Research Centers of America, Hollywood, FL, USA.
✉e-mail: Raffael.Nachbagauer@modernatx.com; Robert.Paris@modernatx.com

(HA), which undergoes continual antigenic change (antigenic drift) that is particularly evident for type A and results in novel variant strains[2,12,13]. Accordingly, the WHO continuously monitors influenza activity and recommends seasonal influenza vaccine compositions twice a year that cover 1 strain for each of the influenza A subtypes (i.e., H1N1 and H3N2) and the influenza B lineages (i.e., Yamagata or Victoria lineage)[2].

Generally, the overall effectiveness of currently available seasonal influenza vaccines, which utilize egg-, cell-, or recombinant protein-based platforms[3], is variable across different populations and seasons. For example, the US Centers for Disease Control and Prevention estimates that overall vaccine effectiveness among the general population ranges between 40 and 60% during seasons when vaccines are well-matched to circulating strains[14]. Notably, in the most recent season (2021–2022), vaccine effectiveness against medically attended acute respiratory infection associated with A/H3N2 (the predominantly circulating strain) was only 16% for individuals of all ages[15]. Several factors can impact effectiveness, including population age, low vaccine-elicited immunogenicity, and strain mismatch[2,13]. Strain mismatch can in part be due to the lengthy production process, resulting in a long lag time from initial strain announcement to vaccine availability, and mutations acquired due to propagation of the virus in eggs or cells (used for the majority of seasonal influenza vaccines)[3,16,17]. Therefore, there remains a public health need for improved seasonal influenza vaccines.

Messenger RNA (mRNA) technology has recently become a promising alternative to conventional vaccine approaches against infectious diseases[18,19]. As demonstrated by mRNA-based vaccines for coronavirus disease 2019 (COVID-19), such as mRNA-1273 (Spikevax®; Moderna, Inc., Cambridge, MA, USA)[20], this technology can enable rapid development of vaccines with a favorable safety profile and high effectiveness against disease[21–24]. An mRNA-based approach to seasonal influenza vaccines could offer several advantages over current methods, including an ability to rapidly respond to strain changes to potentially deploy vaccines more closely matched to circulating strains of a given influenza season and avoidance of egg- and cell-based acquired mutations that can limit vaccine effectiveness.

The investigational mRNA-1010 vaccine is a quadrivalent seasonal influenza vaccine encoding membrane-bound HA surface glycoproteins of four influenza strains (A/H1N1, A/H3N2, B/Victoria, and B/Yamagata) recommended by the WHO for cell- or recombinant vaccines[2,3]. Here, we report interim findings from a first-in-human, phase 1/2 clinical trial of mRNA-1010 in healthy adults aged ≥18 years that evaluated the safety, reactogenicity, and immunogenicity of the vaccine against placebo (Part 1) or a licensed seasonal influenza vaccine as an active comparator (Part 2).

## Results

### Participants

In Part 1 of the study, a total of 180 participants aged ≥18 years were randomly assigned to receive placebo or 1 dose of mRNA-1010 (50 µg, 100 µg, or 200 µg) between July 6, 2021, and August 18, 2021 (Fig. 1). All participants received vaccination; 6 participants subsequently discontinued from the study due to withdrawal of consent or were lost to follow-up. The majority of participants were White and non-Hispanic or Latino; in the mRNA-1010 groups, the median age was 51.0 years and 57.0% were female; in the placebo group, the median age was 50.0 years and 51.1% were female (Supplementary Table S1).

In Part 2 of the trial, a total of 501 participants aged ≥18 years were randomly assigned to receive a single dose of Afluria (n = 53) or mRNA-1010 (25 µg, n = 152; 50 µg, n = 149; 100 µg, n = 147) between November 10, 2021, and November 15, 2021; 498 participants received vaccination (Fig. 1). No participants in the Afluria group discontinued; reasons for study discontinuation in the

mRNA-1010 groups were lost to follow-up (n = 9), withdrawal of consent (n = 5), and death (n = 1; not treatment related (see "Safety")). Most participants in Part 2 were White and non-Hispanic or Latino. In the mRNA-1010 groups, the median age was 52.0 years and 57.3% were female; in the Afluria group, the median age was 53.0 years and 54.7% were female (Supplementary Table S1).

### Safety

In Part 1 of the trial, any solicited local ARs within 7 days of mRNA-1010 vaccination were reported by 82.6%, 85.7%, and 91.3% of participants aged 18–49 years in the 50-µg, 100-µg, and 200-µg groups, respectively, and by 63.6%, 92.0%, and 90.5% of participants aged ≥50 years, respectively (Fig. 2a). Any solicited local ARs after placebo were reported by 20.0% and 20.8% of participants aged 18–49 years or ≥50 years. Grade 3 local ARs were more frequent for participants aged 18–49 years and at the 100-µg and 200-µg mRNA-1010 dose levels. The most common AR was pain at the injection site (Supplementary Fig. S2); this AR was predominantly grade 1, although grade 2 pain was the most common AR after the 200-µg mRNA-1010 dose. Solicited systemic ARs within 7 days of mRNA-1010 vaccination were reported by 78.3%, 90.5%, and 91.3% of participants aged 18–49 years in the 50-µg, 100-µg, or 200-µg groups, respectively, and by 54.5%, 92.0%, and 95.2% of participants aged ≥50 years, respectively (Fig. 2a); 30.0% and 33.3% of participants aged 18–49 and ≥50 years reported a systemic AR after placebo, respectively. The most common systemic ARs were headache, fatigue, myalgia, and chills (Supplementary Fig. S2). Across all mRNA-1010 groups, participants aged 18–49 years more frequently reported grade 3 systemic ARs after vaccination than older participants (13.0–47.8% vs 9.1–23.8%). Overall, grade 3 systemic ARs were more common with higher mRNA-1010 dose levels (100 µg and 200 µg). No grade 4 local or systemic ARs were reported.

In Part 2, any solicited local ARs within 7 days of mRNA-1010 vaccination were reported by 83.3%, 84.2%, and 93.1% of participants aged 18–49 years in the 25-µg, 50-µg, and 100-µg groups, respectively. Among participants aged 50–64 years, 72.4%, 84.5%, and 80.7% of participants reported local ARs for each respective vaccine group, and 67.7%, 83.9%, and 80.0% of participants ≥65 years, respectively, reported local ARs (Fig. 2b). Any solicited local ARs after Afluria were reported by 42.9%, 36.8%, and 30.8% of participants aged 18–49 years, 50 to 64 years, and ≥65 years, respectively. The most common local AR across vaccine groups was pain at the injection site (Supplementary Fig. S3A), which was predominantly grade 1. Across mRNA-1010 vaccine groups, grade 3 local ARs were more frequent in participants aged 18–49 years (0–15.5% participants) than in participants aged ≥50 years; grade 3 local ARs after Afluria were reported by 1 participant (5.3%) aged 50–64 years. In the 25-µg, 50-µg, or 100-µg mRNA-1010 groups, any solicited systemic ARs within 7 days of vaccination were reported by 85.0%, 73.7%, and 91.4%, respectively, of participants aged 18–49 years; 68.4%, 75.9%, and 75.4% of participants aged 50–64 years; and 48.4%, 77.4%, and 76.7% of participants ≥65 years (Fig. 2b). After Afluria, 42.9%, 31.6%, and 38.5% of participants aged 18–49 years, 50 to 64 years, and ≥65 years reported a systemic AR, respectively. Headache, fatigue, myalgia, arthralgia, and chills were the most common systemic ARs (Supplementary Fig. S3). Across mRNA-1010 groups, fever (primarily grade 1 and 2) was reported by 10.2–22.4%, 3.6–8.6%, and 3.2–20.0% of participants aged 18–49 years, 50–64 years, and ≥65 years, respectively. Overall, grade 3 systemic ARs after mRNA-1010 vaccination were reported less frequently for older participants (aged ≥65 years) than those aged 18–49 years and 50–64 years. No grade 4 local or systemic ARs were reported during this part of the study.

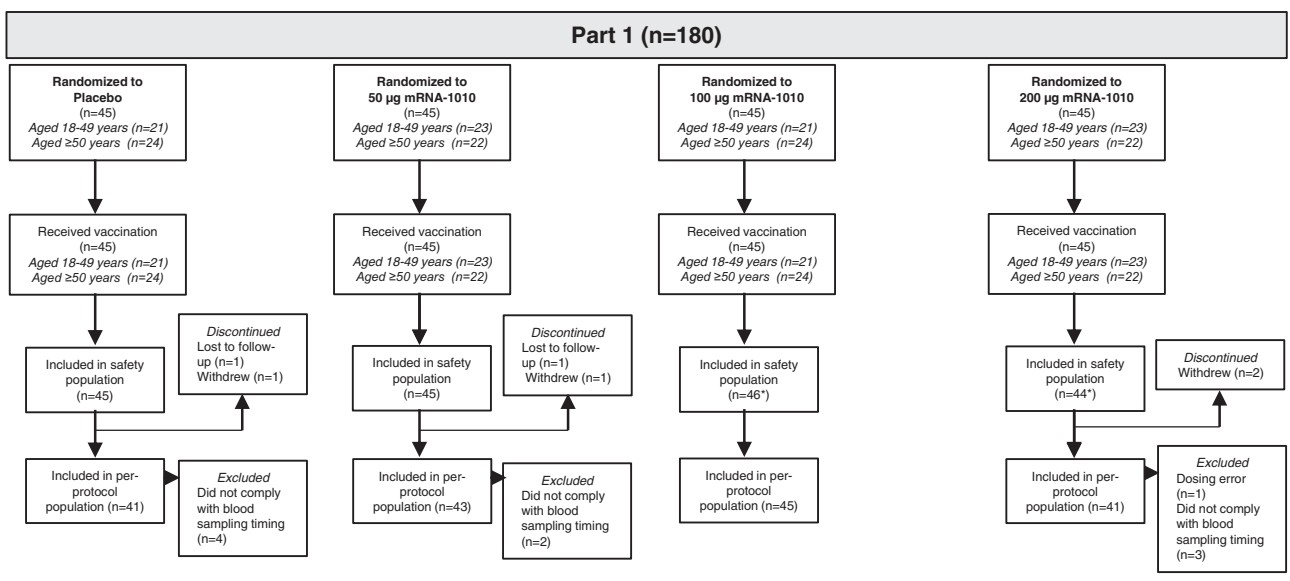

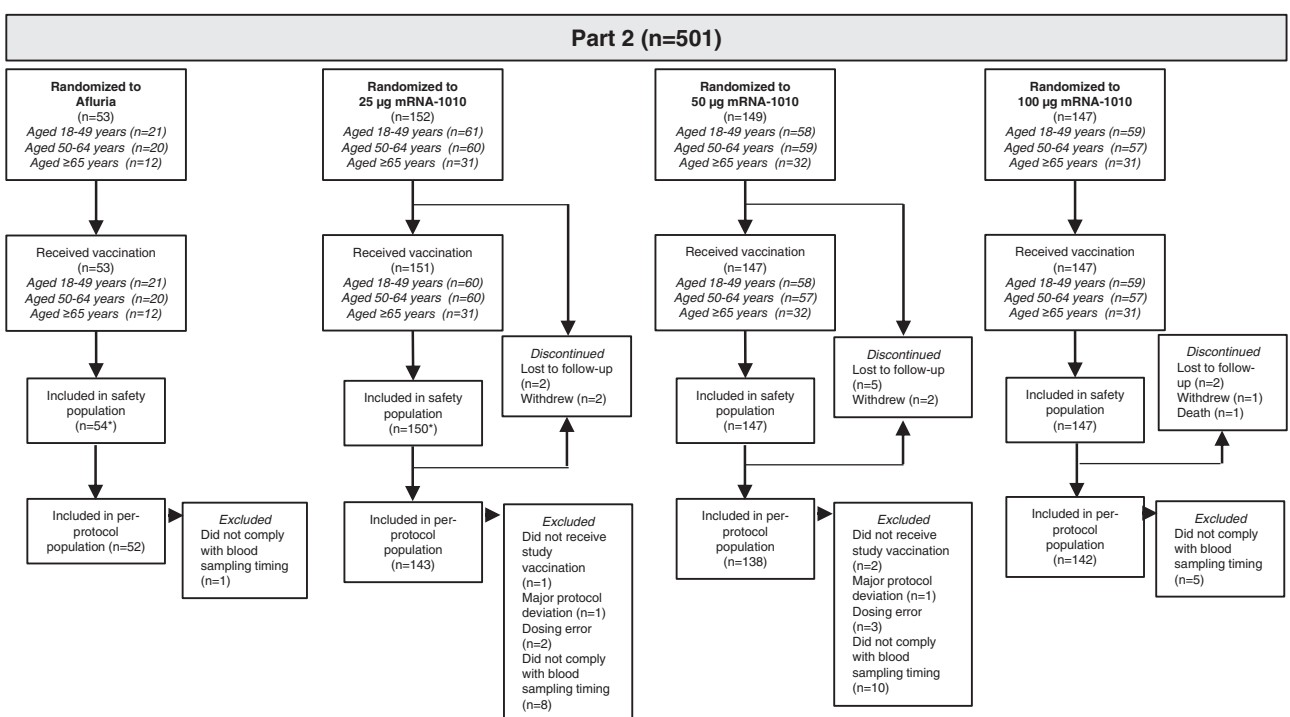

**Fig. 1 | Participant disposition by study part.** One participant in the mRNA-1010 100-µg group in Part 2 of the study died due to cardiac arrest. This adverse event was considered by the investigator as unrelated to the study vaccination. All randomly assigned participants who received study vaccination were included in the safety population; participants were included in the group based on the actual vaccine received. The immunogenicity per-protocol population included all randomly assigned participants who received vaccination and complied with immunogenicity blood sampling timing to have baseline and ≥1 post-vaccination time point assessment, did not have influenza infection at baseline through Day 29 (as documented by polymerase chain reaction), and had no major protocol deviation that impacted the immune response. mRNA messenger RNA. *There was 1 dosing error in Part 1 (*n* = 1 participant randomized to mRNA-1010 200 µg but received 100 µg) and 2 dosing errors in Part 2 (*n* = 1 participant randomized to mRNA-1010 25 µg but received 50 µg; *n* = 1 participant randomized to mRNA-1010 50 µg but received active comparator).

Within 28 days of vaccination in Part 1 of the trial, unsolicited TEAEs were reported by 6 to 7 participants (13.0–15.9%) across each mRNA-1010 group and 6 participants (13.3%) in the placebo group (Table 1). Seven TEAEs were considered treatment-related, including 1 in the placebo group. One participant in the placebo group reported 2 SAEs (acute cystitis and hyponatremia); no SAEs or severe AEs were reported after mRNA-1010 vaccination. MAAEs within 28 days after vaccination were reported by 1 to 3 participants (2.2–6.8%) and 3 participants (6.7%) after mRNA-1010 and placebo, respectively; none were considered related to study vaccination. No deaths or AEs leading to study discontinuation were reported up to Day 29.

In Part 2 of the trial, TEAEs within 28 days of vaccination were reported in 28–41 participants (18.7–27.9%) across mRNA-1010 groups and 8 participants (14.8%) in the Afluria group (Table 1). Twenty-five

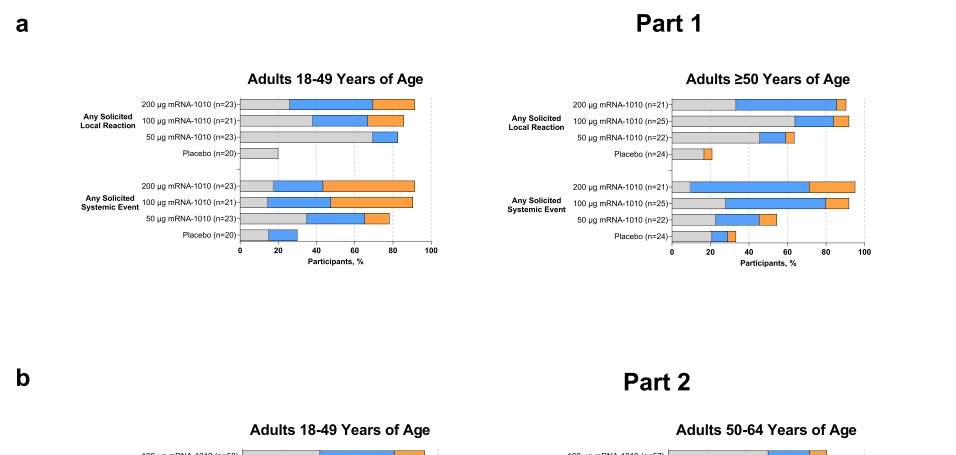

**Fig. 2 | Summary of any solicited local and systemic adverse reactions within 7 days after vaccination by age group in each study part.** Percentages of participants in the solicited safety population reporting any solicited adverse reactions in Part 1 (**a**) or Part 2 (**b**). In Part 1, number of participants in the placebo group were 20 (18–49 years) and 24 (≥50 years); number of participants in the mRNA-1010 groups were 23 (50 μg), 21 (100 μg), and 23 (200 μg) for 18–49 years, and 22, (50 μg), 25 (100 μg), and 21 (200 μg) for ≥50 years. In Part 2, number of participants in the Afluria group were 21 (18–49 years), 19 (50–64 years), and 13 (≥65 years); number of participants in the mRNA-1010 groups were 60 (25 μg), 57 (50 μg), and 58 (100 μg) for 18–49 years; 58 (25 μg), 58 (50 μg), and 57 (100 μg), for 50–64 years; and 31 (25 μg), 31 (50 μg), and 30 (100 μg) for ≥65 years. mRNA, messenger RNA.

(5.6%) and 1 (1.9%) participants reported TEAEs that were considered treatment-related in the mRNA-1010 groups and the Afluria group, respectively. Treatment-related severe unsolicited TEAEs were reported by 2 participants (*n* = 1, mRNA-1010 50 μg (injection site erythema); *n* = 1, mRNA-1010 100 μg (headache)); neither were medically attended. MAAEs were reported by 8–17 participants (5.4–11.6%) and 4 participants (7.4%) in the mRNA-1010 and Afluria groups, respectively; 1 MAAE in the mRNA-1010 25-μg group was considered by the Investigator to be treatment-related (mild upper respiratory tract infection), despite a negative respiratory pathogen polymerase chain reaction panel. Three unsolicited SAEs were reported within 28 days of vaccination in the mRNA-1010 groups; all events were assessed as not related by the Investigator. Two SAEs were reported by participants with a significant medical history of cardiovascular disease (*n* = 1, mRNA-1010 100 μg (cardiac failure congestive in a participant with a history of chronic systolic heart failure and aortic insufficiency/stenosis; event was considered related to medication non-compliance); *n* = 1, mRNA-1010 50 μg (angina pectoris with negative troponins in a participant with a history of coronary artery disease)). The third SAE was a fatal event of unwitnessed cardiac arrest occurring 15 days after vaccination in a 67-year-old male with a medical history of diabetes mellitus, hypertension, and obesity. The participant reported grade 1 injection site pain on Days 2 through 6 after vaccination but did not report any AEs or safety complaints during the scheduled routine safety call at Day 12 after vaccination (3 days prior to his death). This event and the other 2 above-mentioned SAEs were reviewed by the Data Safety Monitoring Board and were considered unrelated to study vaccination. No events of myocarditis or pericarditis were reported.

In Part 1, sporadic grade 1 and 2 abnormalities for chemistry and hematology tests were observed in all groups, but there was no association with mRNA-1010 or dose level. Three grade 3 abnormalities were reported: 1 low hemoglobin level reported in the mRNA-1010 200-μg group, 1 low neutrophil count reported in the mRNA-1010

50-μg group, and 1 elevated alkaline phosphatase reported in the placebo group. No grade 4 laboratory test abnormalities were recorded in Part 1 up to Day 29. No safety laboratory tests were done in Part 2 of the study.

**Immunogenicity**

In Part 1 of the trial, all dose levels of mRNA-1010 (50 μg, 100 μg, and 200 μg) elicited immune responses in participants aged 18–49 years and ≥50 years against all vaccine-matched influenza strains at Day 29 (28 days after vaccination) as measured by HAI (Fig. 3 and Supplementary Table S2). Immune responses after placebo remained close to baseline values. At Day 29, GMTs for all age groups combined exceeded a titer of 1:160 for all strains, which is above the 1:40 level associated with 50% protection against influenza illness[25]. All dose levels of mRNA-1010 generally elicited similar GMTs at Day 29, ranging from 343.8 to 598.2 for A/H1N1, 317.5 to 537.3 for A/H3N2, 173.7 to 281.0 for B/Victoria, and 324.0 to 456.4 for B/Yamagata. From baseline to Day 29, GMFRs for influenza A strains (H1N1 and H3N2) exceeded the 4-fold rise threshold considered for seroconversion in both age groups; however, GMFRs for influenza B strains were lower and did not consistently exceed this 4-fold threshold (GMFRs, B/Victoria: 1.6–2.3 and 1.6–2.1 (18–49 years and ≥50 years, respectively); B/Yamagata: 2.8–3.4 and 3.2–4.9 (18–49 years and ≥50 years, respectively)), which might be driven by high titers at baseline (Fig. 3 and Supplementary Table S2). Seroconversion rates at Day 29 are summarized in Supplementary Table S2.

At 28 days after vaccination (Day 29) in Part 2 of the trial, all dose levels of mRNA-1010 (25 μg, 50 μg, and 100 μg) elicited GMTs for influenza A (H1N1 and H3N2) strains across age groups that exceeded GMTs elicited by Afluria (Fig. 4, Supplementary Table S2, and Supplementary Fig. S4). Among older adults in particular (aged ≥65 years), the ratio of GMTs (95% CIs) for mRNA-1010 versus Afluria ranged from 1.4 (0.6–3.3) to 2.3 (1.0–5.2) for A/H1N1 and 1.6 (0.8–3.2) to 2.9 (1.4–5.7)

**Table 1 | Summary of unsolicited treatment-emergent adverse events through 28 days after vaccination by study part (safety population)**

| | Part 1 | | | | Part 2 | | | |
|---|---|---|---|---|---|---|---|---|
| | Placebo (n = 45) | 50 µg mRNA-1010 (n = 45) | 100 µg mRNA-1010 (n = 46) | 200 µg mRNA-1010 (n = 44) | Afluria (n = 54) | 25 µg mRNA-1010 (n = 150) | 50 µg mRNA-1010 (n = 147) | 100 µg mRNA-1010 (n = 147) |
| **All unsolicited TEAEs, n (%)** | | | | | | | | |
| All | 6 (13.3) | 7 (15.6) | 6 (13.0) | 7 (15.9) | 8 (14.8) | 28 (18.7) | 31 (21.1) | 41 (27.9) |
| SAEs | 1 (2.2) | 0 | 0 | 0 | 0 | 0 | 1 (0.7) | 2 (1.4) |
| Fatal | 0 | 0 | 0 | 0 | 0 | 0 | 0 | 1 (0.7) |
| MAAEs | 3 (6.7) | 1 (2.2) | 2 (4.3) | 3 (6.8) | 4 (7.4) | 12 (8.0) | 8 (5.4) | 17 (11.6) |
| Leading to dose delayed | 0 | 0 | 0 | 0 | 0 | 0 | 0 | 0 |
| Leading to vaccination withdrawn | 0 | 0 | 0 | 0 | 0 | 0 | 0 | 0 |
| Leading to study discontinuation | 0 | 0 | 0 | 0 | 0 | 0 | 0 | 1 (0.7) |
| Severe | 0 | 0 | 0 | 0 | 0 | 1 (0.7) | 1 (0.7) | 3 (2.0) |
| AESIs | 0 | 0 | 0 | 0 | 0 | 0 | 0 | 0 |
| **Related unsolicited TEAEs, n (%)** | | | | | | | | |
| All | 1 (2.2) | 0 | 2 (4.3) | 2 (4.5) | 1 (1.9) | 5 (3.3) | 10 (6.8) | 10 (6.8) |
| SAEs | 0 | 0 | 0 | 0 | 0 | 0 | 0 | 0 |
| Fatal | 0 | 0 | 0 | 0 | 0 | 0 | 0 | 0 |
| MAAEs | 0 | 0 | 0 | 0 | 0 | 1 (0.7) | 0 | 0 |
| Leading to dose delayed | 0 | 0 | 0 | 0 | 0 | 0 | 0 | 0 |
| Leading to vaccination withdrawn | 0 | 0 | 0 | 0 | 0 | 0 | 0 | 0 |
| Leading to study discontinuation | 0 | 0 | 0 | 0 | 0 | 0 | 0 | 0 |
| Severe | 0 | 0 | 0 | 0 | 0 | 0 | 1 (0.7) | 1 (0.7) |
| AESIs | 0 | 0 | 0 | 0 | 0 | 0 | 0 | 0 |

*AE* adverse event, *AESI* adverse event of special interest, *MAAE* medically attended adverse event, *SAE* serious adverse event, *TEAE* treatment-emergent adverse event.
A TEAE was defined as any event not present before study vaccination or any event already present that worsened in intensity or frequency after vaccination.

for A/H3N2 (Fig. 5). For influenza B strains (B/Victoria and B/Yamagata), GMTs at Day 29 across age groups were comparable between all mRNA-1010 dose levels and Afluria. For all age groups combined (Supplementary Fig. S4), GMTs across mRNA-1010 groups ranged from 351.2 to 492.2 for A/H1N1 (Afluria: 186.4), 170.0 to 239.3 for A/H3N2 (Afluria: 101.7), 110.9 to 136.6 for B/Victoria (Afluria: 160.1), and 238.7 to 249.6 for B/Yamagata (Afluria: 204.7); Supplementary Table S2 shows GMTs by each age group. GMFRs from baseline to Day 29 were higher with mRNA-1010 than Afluria for the influenza A strains and were comparable between vaccines for influenza B strains. For all age groups combined (Supplementary Fig. S5), ratios of GMTs (95% CIs) for mRNA-1010 versus Afluria ranged from 1.9 (1.3–2.7) to 2.7 (1.9–3.9) for A/H1N1, 1.8 (1.3–2.5) to 2.6 (1.8–3.6) for A/H3N2, 0.7 (0.5–0.9) to 0.9 (0.7–1.1) for B/Victoria and 1.1 (0.9–1.5) to 1.2 (0.9–1.6) for B/Yamagata. Ratios of GMTs were generally consistent across participants aged 18–49 years, 50–64 years, and ≥65 years (Fig. 5). Seroconversion rates at Day 29 trended higher for the mRNA-1010 groups compared with Afluria for A/H1N1, A/H3N2, and B/Yamagata, while comparable seroconversion rates to Afluria were observed for B/Victoria (Supplementary Table S2).

## Discussion

This manuscript presents interim analysis findings from 2 parts of a phase 1/2, first-in-human clinical trial on the safety and immunogenicity of an investigational mRNA-based quadrivalent vaccine against seasonal influenza (mRNA-1010) in healthy adults ≥18 years. The placebo-controlled Part 1 of this study showed that a single dose of mRNA-1010 (50 µg, 100 µg, or 200 µg) elicited HAI antibodies against vaccine-matched strains at 28 days after vaccination in both younger and older healthy adults. Based on the comparable immunogenicity

across the mRNA-1010 dose levels in Part 1, a lower dose level range of mRNA-1010 (25 µg, 50 µg, or 100 µg) was evaluated in a larger number of participants in Part 2. Findings in Part 2 suggest that mRNA-1010 could elicit higher immunogenicity than a standard-dose influenza vaccine for influenza A strains and comparable immunogenicity for influenza B strains in medically stable adults. Higher responses for influenza A strains remain important, as the A/H3N2 strain in particular causes a larger burden of severe outcomes in older adults[9,10], thus raising the potential for mRNA-1010 to further address this burden in vulnerable age groups.

In both parts of the trial, no treatment-related SAEs were reported nor any safety concerns identified. In Part 1 of the trial, the frequency and severity of solicited ARs generally increased in a dose-dependent manner and was higher among younger (18–49 years) than older (≥50 years) adults. In Part 2, all three mRNA-1010 dose levels had an acceptable reactogenicity profile. For all age groups, solicited ARs were more common with mRNA-1010 than Afluria and were typically grade 1 or grade 2 in severity. No grade 4 solicited ARs were reported for any vaccine group and grade 3 events were less frequent for lower dose levels and for older adults (≥65 years). Overall, these safety data support the continued evaluation of mRNA-1010.

A single dose of mRNA-1010 (50 µg, 100 µg, or 200 µg) in Part 1 of the trial elicited HAI antibodies against all vaccine-included strains at Day 29 in adults irrespective of participant age. Overall, the 50-µg dose level induced HAI titers that were comparable to those elicited by higher dose levels (100 µg and 200 µg). In both younger and older adults, the GMFRs from baseline exceeded 4-fold for influenza A strains, which is notable, as influenza A strains are the primary drivers of influenza-related hospitalizations and deaths, particularly in older adults[9]. Although the 4-fold

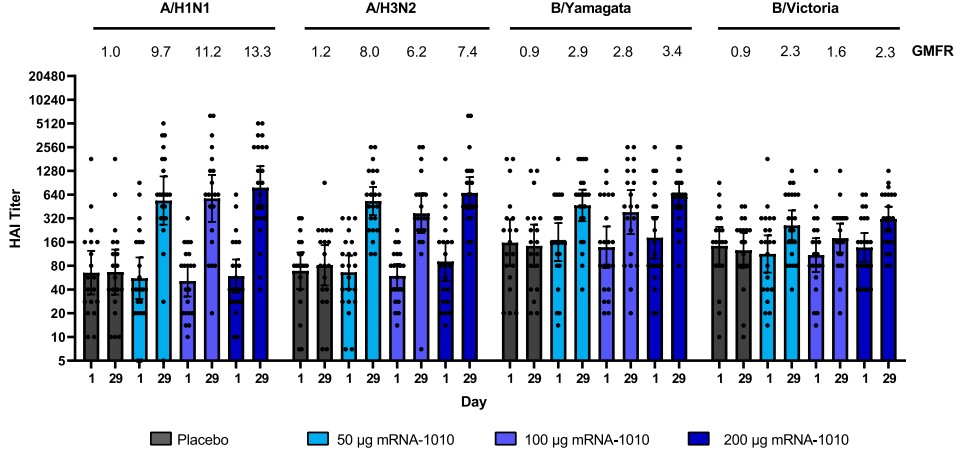

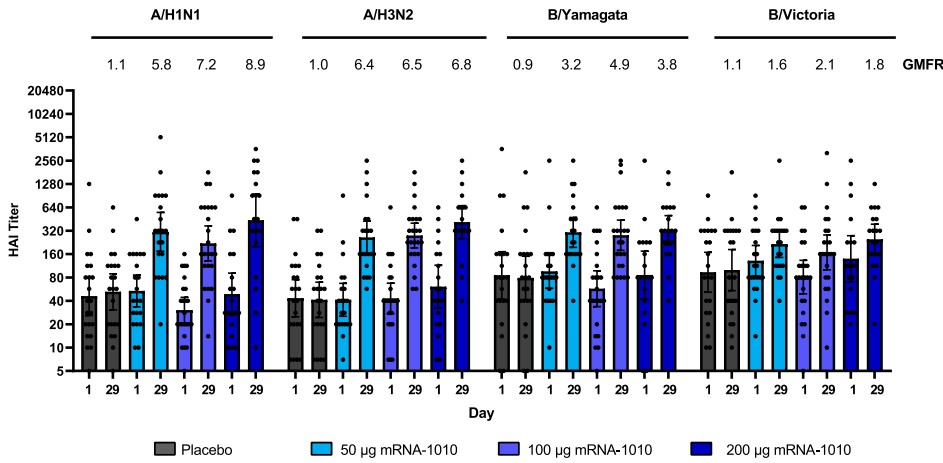

**Fig. 3 | GMTs and GMFRs of anti-hemagglutinin antibodies for vaccine-matched seasonal influenza strains in adults in Part 1.** Hemagglutination inhibition GMTs with associated 95% CIs against seasonal influenza A strains (A/Wisconsin/588/2019[H1N1]pdm09 and A/Hong Kong/45/2019[H3N2]) or influenza B strains (B/Washington/02/2019 (B/Victoria lineage) and B/Phuket/3073/2013 (B/Yamagata lineage)) are shown at Day 1 (baseline) and Day 29 (28 days after vaccination) among participants **a** 18–49 years and **b** ≥50 years of age in the per-protocol population for Part 1. Dots correspond to participant-level titers (18–49 years, *n* = 19 (placebo), 22 (mRNA-1010 50 µg), 21 (mRNA-1010 100 µg), and 22 (mRNA-1010 200 µg); ≥50 years, *n* = 22 (placebo), 21 (mRNA-1010 50 µg), 24 (mRNA-1010 100 µg), and 19 (mRNA-1010 200 µg). GMFRs at Day 29 from Day 1 are shown above each Day 29 bar plot. LLOQs were 10 (H1N1, B/Victoria, and B/Yamagata) or 14 (H3N2); ULOQs were 6400 (H1N1, H3N2, and B/Yamagata) or 3200 (B/Victoria). HAI GMTs <LLOQ were replaced by 0.5× LLOQ and values > ULOQ were converted to ULOQ. CI confidence interval, HAI hemagglutination inhibition, GMFR geometric mean fold-rise, GMT geometric mean titer, LLOQ lower limit of quantification, mRNA messenger RNA, ULOQ upper limit of quantification.

threshold was not consistently met for influenza B strains, these findings could reflect high baseline titers; however, lower influenza B strain responses have also been observed for other licensed influenza vaccines[1]. Immunogenicity findings from Part 2 of the trial showed that all dose levels of mRNA-1010 (25 µg, 50 µg, or 100 µg) elicited high levels of HAI antibodies on Day 29, exceeding the 1:40 threshold associated with a 50% reduced risk of infection. Overall, regardless of participant age, all dose levels of mRNA-1010 induced functional antibody responses that were higher than Afluria for influenza A strains and were similar for influenza B strains. Notably, GMT ratios for mRNA-1010 versus Afluria were generally in line with those from a clinical trial comparing enhanced influenza vaccines to a standard dose vaccine in older adults[1]. Taken together, these preliminary immunogenicity findings suggest that mRNA-1010 has the potential to further address the high burden of influenza, particularly in older adults.

Overall, these first-in-human safety and immunogenicity findings in adults aged ≥18 years support the continued investigation of mRNA-1010 against seasonal influenza and highlight the potential of the mRNA platform to improve the effectiveness of influenza vaccines. Vaccines using mRNA technology are readily amenable to both antigenic drift and shift in influenza strains, allowing for rapid deployment of vaccines that are more closely matched to currently circulating strains and can avoid aberrant mutations in vaccine antigens caused by egg- or cell-culture approaches[26,27]. In addition, mRNA-based platforms can allow for expression of multiple antigens, raising the possibility for

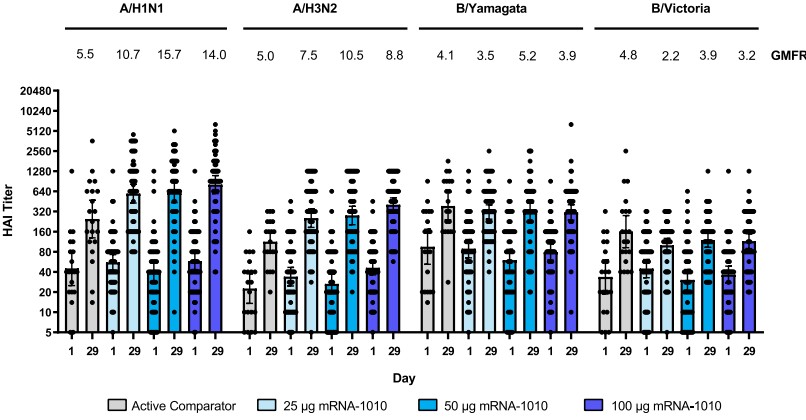

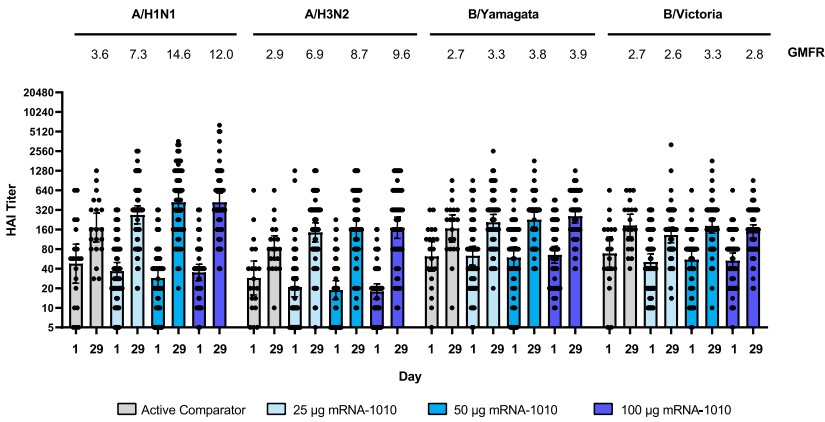

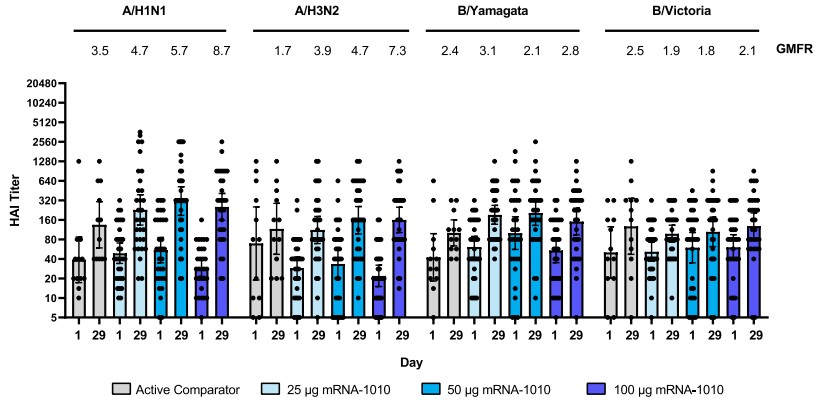

increased breadth of protective responses against seasonal influenza or against multiple respiratory diseases. Further, based on findings with mRNA-1273, an mRNA-based vaccine against SARS-CoV-2, mRNA vaccines may also induce strong cellular immune responses and prolonged germinal center reactions[28–31] that can improve protection in older adults, a population at particular risk for infection and severe outcomes[10]. Further laboratory assessments to evaluate these potential advantages are ongoing. While mRNA-1010 had an acceptable safety profile in this trial, transient solicited ARs were more common after mRNA-1010 than with the active comparator. Additional clinical trials are ongoing to further assess the safety, efficacy, and immunogenicity of this vaccine and a licensed influenza vaccine comparator (NCT05566639 and NCT05415462).

**Fig. 4 | GMTs and GMFRs of anti-hemagglutinin antibodies for vaccine-matched seasonal influenza strains in adults in Part 2.** Hemagglutination inhibition GMTs with associated 95% CIs against seasonal influenza A strains (A/Wisconsin/588/2019[H1N1]pdm09 and A/Cambodia/e0826360/2020[H3N2]) or influenza B strains (B/Washington/02/2019 (B/Victoria lineage) and B/Phuket/3073/2013 (B/Yamagata lineage)) are shown at Day 1 (baseline) and Day 29 (28 days after vaccination) among participants **a** 18–49 years, **b** 50–64 years, and **c** ≥65 years of age in the per-protocol population for Part 2. Dots corresponding to participant-level titers; numbers of participants in the Afluria groups were 20 (18–49 years), 20 (50–64 years), and 12 (≥65 years); numbers of participants in the mRNA-1010 groups were 55 (25 μg), 53 (50 μg), and 58 (100 μg) for 18–49 years; 57 (25 μg), 54 (50 μg), and 54 (100 μg) for 50–64 years; and 31 (25 μg), 31, (50 μg), and 30 (100 μg) for ≥65 years. GMFRs at Day 29 from Day 1 are shown above each Day 29 bar plot. LLOQs were 10 for each influenza strain; ULOQs were 6400 (H1N1 and B/Yamagata), 1280 (H3N2), or 3200 (B/Victoria). HAI GMTs <LLOQ were replaced by 0.5× LLOQ and values >ULOQ were converted to ULOQ. CI confidence interval, HAI hemagglutination inhibition, GMFR geometric mean fold-rise, GMT geometric mean titer, LLOQ lower limit of quantification, mRNA messenger RNA, ULOQ upper limit of quantification.

Study strengths include the randomized, observer-blind, placebo-controlled (Part 1) and active comparator (Part 2) design. The study design did not specify any hypothesis testing; thus, the sample sizes of participants enrolled (Part 1, $N = 180$; Part 2, $N = 501$) restricted statistical comparisons across groups. Potential limitations of the study include the enrollment of higher percentages of White, non-Hispanic or Latino participants that may limit generalizability to other demographics. Additional information on the safety and immunogenicity of mRNA-1010 will be gained from the final analyses of these 2 parts of the phase 1/2 trial, which will assess the longevity of immune responses through Day 181, as well as the phase 2 extension of this trial that will evaluate lower dose levels of mRNA-1010 in adults. Subsequent analyses will also include assessments of HA-specific T-cell and B-cell responses. Larger phase 3 trials of mRNA-1010 safety and immunogenicity in ~6000 adults aged ≥18 years (NCT05415462) as well as the safety and efficacy of mRNA-1010 in approximately 23000 adults aged ≥50 years (NCT05566639) are also currently underway. Further studies will also evaluate mRNA-based seasonal influenza vaccines against enhanced influenza vaccines (ie, high-dose, recombinant protein, or adjuvanted vaccines) in adult populations (NCT05397223, NCT05606965, and NCT05333289).

In conclusion, the interim analysis findings from the placebo-controlled Part 1 and active comparator controlled Part 2 of this phase 1/2 trial of mRNA-1010 in adults aged ≥18 years raised no safety concerns and showed that the vaccine was immunogenic against all tested influenza strains in both younger and older adults. Further, the investigational mRNA-1010 vaccine elicited either higher or comparable immune responses to a standard-dose, quadrivalent inactivated vaccine. These findings support the continued clinical development of mRNA-1010 to combat seasonal influenza.

## Methods

### Trial design and participants
This is a first-in-human, phase 1/2, randomized, observer-blinded study at 20 sites in the United States to evaluate the safety, reactogenicity, and immunogenicity of mRNA-1010 in adults ≥18 years of age (NCT04956575). The study comprises 3 parts, which assessed mRNA-1010 or placebo in healthy adults in Part 1, followed by additional dose-ranging assessments in Parts 2 and 3 to evaluate mRNA-1010 versus licensed comparator vaccines in medically stable adults. This report summarizes interim findings for Parts 1 and 2 of the study, with final results to be separately reported.

Eligible participants in Part 1 of the study were healthy adults ≥18 years of age (Supplementary Fig. S1). A full list of inclusion and exclusion criteria, as well as further design details for each part of this study, are included in the Supplementary Methods as well as the Trial Protocol and Statistical Analysis Plan within the Supplementary Information. The initial stage of Part 1 planned for ~36 participants (9 participants in each group) to be randomly assigned (1:1:1:1) to receive a single dose of mRNA-1010 (50 μg, 100 μg, or 200 μg) or placebo. The randomized allocation schedule was generated by the sponsor's biostatistics department or designee. Safety data up to 7 days after

vaccination were reviewed by a blinded internal safety team; after confirmation that no study pause rules were met, the remaining participants (~36 in each group for a total of 144 participants) were randomly assigned in the Part 1 expansion stage. Randomization at this stage was stratified by age (18–49 years and ≥50 years) and was balanced within each group. Strain selection for mRNA-1010 in Part 1 was based on WHO recommendations for the 2021 Southern Hemisphere (SH) vaccine composition.

Part 2 of the study enrolled medically stable adults ≥18 years of age, excluding those adults with chronic diseases requiring ongoing medical intervention within the 3 months before enrollment and with immunocompromising conditions or medications. Approximately 500 participants were planned to be randomly assigned (3:3:3:1) to receive a single dose of mRNA-1010 (25 μg, 50 μg, or 100 μg) or a licensed quadrivalent seasonal influenza vaccine (Afluria®; Seqirus Pty Ltd, Parkville, Victoria, Australia; Supplementary Fig. S1). The sponsor's biostatistics department or designee generated the randomized allocation schedule for vaccine assignment. Randomization was performed in parallel among the 4 vaccine groups, and participants were stratified by age (18–49 years, 50–64 years, or ≥65 years) and vaccination status in the previous influenza season (received or not received). As vaccinations during Part 2 of the study were planned during the NH influenza season, strain selection for mRNA-1010 in Part 2 was based on WHO recommendations for the 2021/2022 NH vaccine composition.

In each part of the study, participants were followed for approximately 6 months, with a single dose of the vaccine administered on Day 1 and the final study visit on Day 181 (Month 6). Both parts of the study had a planned interim analysis to evaluate safety and immunogenicity data of all participants through Day 29. Participants in Part 1 were recommended to receive a licensed 2021–2022 NH seasonal influenza vaccine after Day 29 of the study.

The study was conducted in accordance with the protocol, applicable laws, and regulatory requirements, as well as International Council for Harmonization Good Clinical Practice guidelines, and the consensus ethical principles derived from international guidelines, including the Declaration of Helsinki and Council for International Organizations of Medical Sciences International Ethical Guidelines. The protocol was approved by the central institutional review board (Advarra, Inc., Columbia, MD) prior to study initiation, and written informed consent was obtained from all participants before enrollment.

### Vaccines
In Part 1, participants received mRNA-1010 or placebo (normal saline), while in Part 2, participants received mRNA-1010 or a licensed standard dose quadrivalent influenza vaccine (Afluria). mRNA-1010 includes mRNAs encoding for the surface glycoprotein HA of four influenza virus strains formulated in lipid nanoparticles. In Part 1, mRNA-1010 encoded influenza strains recommended by the WHO for 2020-2021 SH cell- or recombinant-based vaccines (see Supplementary Methods for details). In Part 2, mRNA-1010 encoded strains recommended by the WHO for 2021–2022 NH cell- or recombinant-based vaccines (see

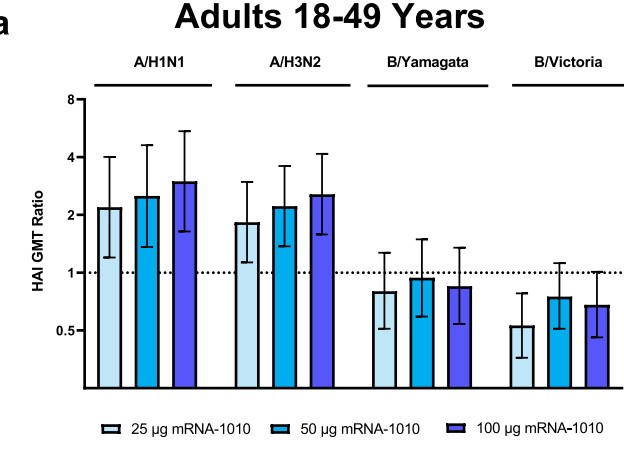

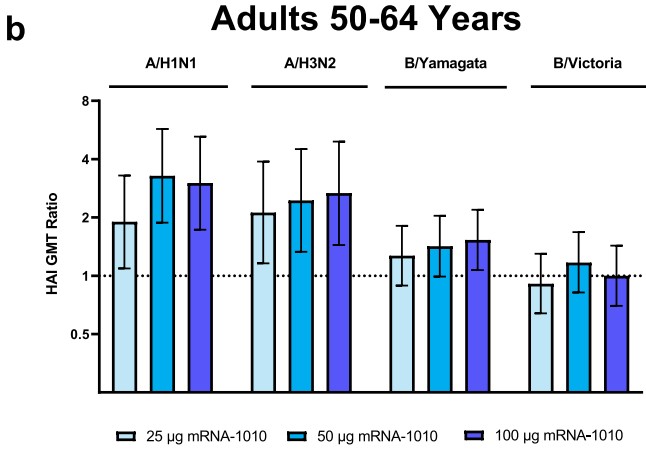

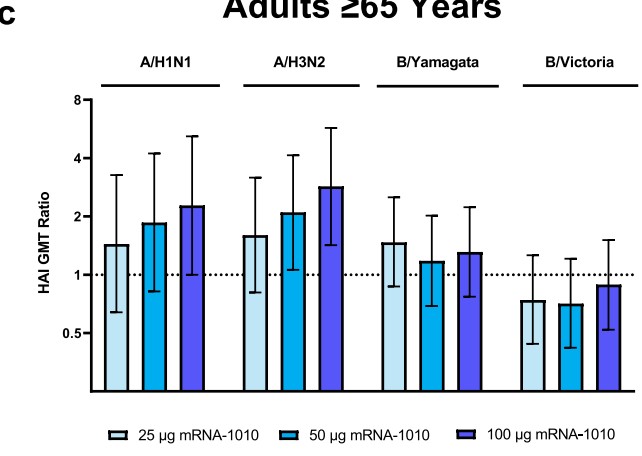

**Fig. 5 | Ratios of GMTs of anti-hemagglutinin antibodies after vaccination with mRNA-1010 compared with Afluria in adults in Part 2.** Ratios of HAI GMTs with associated 95% CIs against vaccine-matched seasonal influenza strains (A/Wisconsin/588/2019[H1N1]pdm09, A/Cambodia/e0826360/2020[H3N2], B/Washington/02/2019 (B/Victoria lineage), and B/Phuket/3073/2013 (B/Yamagata lineage)) at 28 days after vaccination with mRNA-1010 compared with Afluria are shown for participants **a** 18–49 years, **b** 50–64 years, and **c** ≥65 years of age in the per-protocol population for Part 2. Horizontal dotted line indicates a GMT ratio of 1, which reflects comparable GMTs between mRNA-1010 and Afluria. Numbers of participants in the Alfuria groups were 20 (18–49 years), 20 (50–64 years), and 12 (≥65 years); numbers of participants in the mRNA-1010 groups were 55 (25 μg), 53 (50 μg), and 58 (100 μg) for 18–49 years; 57 (25 μg), 54 (50 μg), and 54 (200 μg) for 50–64 years; and 31 (25 μg), 31 (50 μg), and 30 (100 μg) for ≥65 years. CI confidence interval, HAI hemagglutination inhibition, GMT geometric mean titer, LLOQ lower limit of quantification, mRNA messenger RNA, ULOQ upper limit of quantification.

and to evaluate the humoral immunogenicity of mRNA-1010 and active comparator against vaccine-matched influenza A and B strains at Day 29 (primary and secondary).

### Safety assessments

Safety endpoints included solicited local and systemic adverse reactions (ARs) for 7 days after vaccination, safety laboratory abnormalities (Part 1 only), unsolicited adverse events (AEs) for 28 days after vaccination, as well as serious AEs (SAEs), AEs of special interest (AESIs), and medically attended AEs (MAAEs) through the end of the study (Day 181). Interim safety data through Day 29 are included in this report. Participants used an electronic diary to record local ARs (ie, injection site pain, injection site redness, injection site hardness, or axillary swelling/tenderness ipsilateral to the side of injection), or systemic ARs (ie, headache, fatigue, myalgia, arthralgia, nausea/vomiting, chills, and fever). Safety laboratory assessments in Part 1 (at baseline and Day 8) included white blood cell count, hemoglobin, platelets, alanine aminotransferase, aspartate aminotransferase, total bilirubin, alkaline phosphatase, and creatinine.

### Immunogenicity assessments

Blood samples for immunogenicity assessments were collected on Days 1 (baseline), 8 (Part 1 only), 29, and 181 (end of study). This report summarizes immunogenicity assessments at baseline and Day 29. Immunogenicity endpoints included geometric mean titers (GMTs) at Day 1 and 29, geometric mean fold rises (GMFRs) at Day 29 over Day 1 (baseline), and percentage of participants with seroconversion at Day 29 of serum anti-HA antibodies against vaccine-matched influenza A and B strains as measured by hemagglutination inhibition (HAI) assay using red blood cells from guinea pig and cell-grown viruses as described in the Supplementary Methods.

### Statistical analyses

This phase 1/2 study did not test any formal statistical hypotheses. Sample size (described in the Supplementary Methods) was considered sufficient to provide a descriptive summary of the safety and immunogenicity of different dose levels of mRNA-1010. All safety assessments except for solicited local and systemic ARs were assessed in the safety population, which included all randomized participants who received vaccination. Solicited ARs were assessed in all participants in the safety population who contributed any solicited AR data (solicited safety population). The number of events of unsolicited AEs, SAEs, AESIs, and MAAEs were summarized, while descriptive summary statistics were provided for all other safety analyses.

Immunogenicity analyses were performed in the per-protocol population, which included all randomly assigned participants who received vaccination and complied with immunogenicity blood

Supplementary Methods for details). mRNA-1010 was provided as a sterile liquid for injection and diluted to different dose levels with normal saline. All vaccines were administered intramuscularly as a single 0.5-mL injection.

### Objectives

The primary objectives of Part 1 were to evaluate the safety and reactogenicity of a single dose of mRNA-1010 (50 μg, 100 μg, and 200 μg) versus placebo and to evaluate the humoral immunogenicity of a single dose of mRNA-1010 against vaccine-matched influenza A and B strains at Day 29. The objectives of Part 2 were to evaluate the safety and reactogenicity of mRNA-1010 (25 μg, 50 μg, and 100 μg; primary)

sampling timing to have baseline and ≥1 post-vaccination time point assessment, did not have influenza infection at baseline through Day 29 (as documented by polymerase chain reaction), and had no major protocol deviation that impacted the immune response. The geometric mean of specific antibody titers with corresponding 95% confidence intervals (CIs) at Day 29 and GMFRs of specific antibody titers with corresponding 95% CI at Day 29 over Day 1 (baseline) were calculated for each vaccination group; 95% CIs were calculated based on the $t$ distribution of the log 2-transformed values and then back transformed to the original scale. The seroconversion rate from baseline was determined along with two-sided 95% CIs using the Clopper–Pearson method. The rate of seroconversion was defined as the percentage of participants with either a prevaccination HAI titer <1:10 and a postvaccination HAI titer ≥1:40 or a prevaccination HAI titer ≥1:10 and a minimum fourfold rise in postvaccination HAI antibody titer. In Part 2, GMTs and seroconversion rates at Day 29 were compared in the mRNA-1010 groups with the active comparator group (see Supplement for details). Statistical analyses were performed using SAS version 9.4.

### Reporting summary
Further information on research design is available in the Nature Portfolio Reporting Summary linked to this article.

## Data availability
Access to patient-level data presented in this article (antibody assays, safety, and reactogenicity) and supporting clinical documents with external researchers who provide methodologically sound scientific proposals will be available upon reasonable request and subject to review from 2 years after study completion. Such requests can be made to Moderna Inc., 200 Technology Square, Cambridge, MA 02139. A materials transfer and/or data access agreement with the sponsor will be required for accessing shared data. All other relevant data are presented in the paper. The protocol is available as online supplementary material to this article. ClinicalTrials.gov: NCT04956575.

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

## Acknowledgements
We thank the participants for their dedication and contributions to the study, the Study Investigators, and our clinical team colleagues at PPD for their collaboration. Medical writing and editorial assistance were provided by Emily Stackpole, PhD, and Jared MacKenzie, PhD, of

MEDiSTRAVA in accordance with Good Publication Practice (GPP3) guidelines, funded by Moderna, Inc., and under the direction of the authors. Moderna, Inc., was involved in the study design, data collection and analysis, and the writing of this manuscript. This study was funded by Moderna, Inc.

## Author contributions

I.L., R.N., L.C., K.S., D.S., C.H., W.H., D.R.S., J.A., and R.P. contributed to the study concept and design. Data were collected by DE and HS, then analyzed and interpreted by I.L., R.N., D.E., H.S., A.A., R.C., W.H., D.R.S., J.A., and R.P. All authors contributed to the drafting and critical review of this manuscript and approved the final draft.

## Competing interests

I.L., R.N., L.C., K.S., A.A., D.S., C.H., R.C., W.H., D.R.S., J.A., and R.P. are employees of and shareholders in Moderna, Inc. D.E. and H.S. declare no competing interests.
