## [Peer Review File · Nature Communications]

Safety and Immunogenicity of a Phase 1/2 Randomized Clinical Trial of a Quadrivalent, mRNA-based Seasonal Influenza Vaccine (mRNA-1010) in Healthy Adults: Interim AnalysisReviewers' Comments:

Reviewer #1:

Remarks to the Author:

This paper reports encouraging results from a phase 2 study evaluating a quadrivalent mRNA based influenza vaccine. The study results are straightforward, but would consider a few minor alterations to the manuscript

Abstract. If the abstract will be making the case that the immunogenicity of the mRNA vaccine was greater than the comparator vaccine for the influenza A viruses, it should also be mentioned that the reactogenicity of the mRNA vaccine was also substantially greater than that of the comparator inactivated vaccine.

Introduction (line 68): The authors note that passage of virus in eggs or MDCK cells can result in HA mutations related to adaptation to the receptors. Reference 16 addresses egg adaptations directly, but reference 3 is a review article (a very nice one) that doesn't really discuss cell culture adaptation. The authors could consider citing a different paper from the same lab (Takada Nature Microbiology 4:1268-1273, 2019) that addresses this more directly.

Adverse events (line 163) it might be relevant to mention whether there was any workup among the 3 cardiac SAEs for evidence of myocarditis following vaccination, and/or possibly how potential myocarditis was assessed in the study.

Discussion. In discussing the potential limitations (line 262 and following) the authors should include among the limitations that the study does not assess the longevity of the antibody response, as only titers on day 28 were reported. If there are plans for longer term immunogenicity assessments (e.g., day 91), this might be mentioned here.

Reviewer #2:

Remarks to the Author:

Lee and colleagues conducted a randomized, observer-blind, controlled trial comparing influenza mRNA vaccine with saline (part1) and inactivated influenza vaccine (part2). The authors reported no safety concerns and higher peak HAI titers of the mRNA vaccine. The Methods and Results are in general well written. I have no comment on the study design, data presenting but have some comments on discussion and conclusion as below. It would be beneficial for the authors to address these comments in order to provide a more accurate interpretation of the study results.

Discussion: In the first paragraph, the authors mentioned that H3N2 caused a larger burden of severe outcomes in older adults, and emphasized the importance of higher responses against flu A strains. However, the data in Supplement Table 2, specifically the H3N2 Hemagglutination Inhibition (HAI) at D29 among 65 years or older, does not appear to clearly demonstrate a significant difference between the Afluria, 25, 50, and 100 microgram mRNA vaccine recipients. It would be beneficial for the authors to further interpret this, or provide additional data or analysis to support their conclusion that mRNA-1010 offers higher immune responses/protection against H3N2 in older adults. Additionally, is there a significant difference by groups? Any tests done? Without this additional information, it may be difficult for readers to fully understand the significance of the data presented in Supplement Table 2 and the conclusion drawn by the authors.

Besides, I would like to see some discussions on the AEs, in particular among elderly adults of at least 65 years. Considering the systemic solicited events in that age group is much higher than inactivated vaccine, with around 50% middle or high dose groups having grade 2 or grade 3 AE, whilst I can't see any grade 2 or grade 3 AE in the inactivated vaccine group. This raises concerns about the risk-benefit ratio of mRNA-1010 in older adults, particularly given that the HAI titers for this age group against H3N2 and flu B are similar to those of the inactivated vaccine group. The higher proportion of systemic

AEs and similar antibody responses suggest that the inactivated vaccine may be a better alternative for older adults. The authors should provide a discussion on the safety profile of mRNA-1010 in older adults and the potential trade-offs between the higher proportions of AEs and the similar or marginally higher immune responses of mRNA-1010 across different doses. Providing the study protocol and statistical analysis plan as supplementary materials would help to increase the transparency and rigor of the study, and would allow readers to better evaluate the validity and reliability of the results.

Reviewer #3:

Remarks to the Author:

This manuscript describes a first-in-human Phase 1/2 dose-finding trial of a quadrivalent influenza mRNA vaccine manufactured using a platform that has demonstrated good efficacy against the SARS-CoV-2 virus with an acceptable safety profile. The Phase 1 portion of the current study appears to have been conducted in Australia in 2021 and the Phase 2 in the US in 2021-2022. The dates of enrollment would be important to allow assessment of the role of recent exposure to wild-type virus, especially the B lineages.

Unfortunately, the sample sizes of the two portions of the study are insufficient to support a clear distinction with respect to optimal dose (except as the higher doses are clearly associated with more frequent and severe reactogenicity, especially in younger adults). Furthermore, the use of an egg-based vaccine as the active comparator does not allow the assessment of relative immunogenicity between the mRNA platform and the more relevant comparator of a recombinant platform.

Of note, Figure 5 does not appear to present the immunogenicity induced by the comparator, despite the title of the figure - an apparent oversight.

In summary, the study certainly supports the conclusion that further evaluation is warranted. The magnitude and severity of reactogenicity may prove to be a challenge to annual repeated vaccination.

Reviewer 1

This paper reports encouraging results from a phase 2 study evaluating a quadrivalent mRNA based influenza vaccine. The study results are straightforward, but would consider a few minor alterations to the manuscript

Comment 1. Abstract. If the abstract will be making the case that the immunogenicity of the mRNA vaccine was greater than the comparator vaccine for the influenza A viruses, it should also be mentioned that the reactogenicity of the mRNA vaccine was also substantially greater than that of the comparator inactivated vaccine.

Author response and action taken: We agree with the Reviewer and have revised the Abstract to now state *"No safety concerns were identified; solicited adverse reactions were dose-dependent and more frequent after receipt of mRNA-1010 than the active comparator."*

Comment 2. Introduction (line 68): The authors note that passage of virus in eggs or MDCK cells can result in HA mutations related to adaptation to the receptors. Reference 16 addresses egg adaptations directly, but reference 3 is a review article (a very nice one) that doesn't really discuss cell culture adaptation. The authors could consider citing a different paper from the same lab (Takada Nature Microbiology 4:1268-1273, 2019) that addresses this more directly.

Author response and action taken: We thank the Reviewer for their suggestion and have accordingly cited the indicated reference to support this sentence in the Introduction.

Comment 3. Adverse events (line 163) it might be relevant to mention whether there was any workup among the 3 cardiac SAEs for evidence of myocarditis following vaccination, and/or possibly how potential myocarditis was assessed in the study.

Author response and action taken: The manuscript has been revised to explicitly state that no events of myocarditis or pericarditis were reported in this study. Further, the Results section of the manuscript now provides further context on the 3 cardiac SAEs, as follows:

“Two SAEs were reported by participants with a significant medical history of cardiovascular disease (n=1, mRNA-1010 100 µg [cardiac failure congestive in a participant with a history of chronic systolic heart failure and aortic insufficiency/stenosis; event was considered related to medication non-compliance]; n=1, mRNA-1010 50 µg [angina pectoris with negative troponins in a participant with a history of coronary artery disease]). The third SAE was a fatal event of unwitnessed cardiac arrest occurring 15 days after vaccination in a 67-year-old male with a medical history of diabetes mellitus, hypertension, and obesity. The participant reported grade 1 injection site pain on Days 2 through 6 after vaccination but did not report any AEs or safety complaints during the scheduled routine safety call at Day 12 after vaccination (3 days prior to his death). This event and the other 2 above-mentioned SAEs were reviewed by the Data Safety Monitoring Board and were considered unrelated to study vaccination. No events of myocarditis or pericarditis were reported.”

Comment 4. Discussion. In discussing the potential limitations (line 262 and following) the authors should include among the limitations that the study does not assess the longevity of the antibody response, as only titers on day 28 were reported. If there are plans for longer term immunogenicity assessments (e.g., day 91), this might be mentioned here.

Author response and action taken: The limitations paragraph of the Discussion has been revised to note that the duration of antibody responses through Day 181 (Month 6) are planned to be evaluated in this study and are forthcoming, as follows: *“Additional information on the safety and immunogenicity of mRNA-1010 will be gained from the final analyses of these 2 parts of the phase 1/2 trial, which will assess the longevity of immune responses through Day 181, as well as the phase 2 extension of this trial that will evaluate lower dose levels of mRNA-1010 in adults.”*

Reviewer 2

Lee and colleagues conducted a randomized, observer-blind, controlled trial comparing influenza mRNA vaccine with saline (part1) and inactivated influenza vaccine (part2). The authors reported no safety concerns and higher peak HAI titers of the mRNA vaccine. The Methods and Results are in general well written. I have no comment on the study design, data presenting but have some comments on discussion and conclusion as below. It would be beneficial for the authors to address these comments in order to provide a more accurate interpretation of the study results.

Comment 1. Discussion: In the first paragraph, the authors mentioned that H3N2 caused a larger burden of severe outcomes in older adults, and emphasized the importance of higher responses against flu A strains. However, the data in Supplement Table 2, specifically the H3N2 Hemagglutination Inhibition (HAI) at D29 among 65 years or older, does not appear to clearly demonstrate a significant difference between the Afluria, 25, 50, and 100 microgram mRNA vaccine recipients. It would be beneficial for the authors to further interpret this, or provide additional data or analysis to support their conclusion that mRNA-1010 offers higher immune responses/protection against H3N2 in older adults. Additionally, is there a significant difference by groups? Any tests done? Without this additional information, it may be difficult for readers to fully understand the significance of the data presented in Supplement Table 2 and the conclusion drawn by the authors.

Author response and action taken: We appreciate the Reviewer's comment and note that due to small samples sizes, no statistical comparisons between groups were performed in this phase 1/2 study, as currently stated in the Discussion, as follows: *"The study design did not specify any hypothesis testing; thus, the sample sizes of participants enrolled (Part 1, N=180; Part 2, N=501) restricted statistical comparisons across groups."* Although these statistical comparisons could not be performed, we note that the ratio of GMTs for mRNA-1010 versus Afluria at Day 29 among older adults (≥ 65 years) were 1.60 (25 μg), 2.10 (50 μg) and 2.86 (100 μg).

While not specific to adults ≥ 65 years, to increase the sample size and more clearly show differences between groups we have now also revised the manuscript to include 2 supplemental figures that graphically display immunogenicity findings for all age groups combined (all adults ≥ 18 years of age). Supplement Figure 4 shows GMTs and GMFRs for each influenza strain, while

Supplement Figure 5 shows the ratio of GMTs for mRNA-1010 versus Afluria at Day 29 for all age groups combined. Together, these data more clearly show the increased titers against A/H3N2 elicited by mRNA-1010 versus Afluria at Day 29.

Comment 2. Besides, I would like to see some discussions on the AEs, in particular among elderly adults of at least 65 years. Considering the systemic solicited events in that age group is much higher than inactivated vaccine, with around 50% middle or high dose groups having grade 2 or grade 3 AE, whilst I can't see any grade 2 or grade 3 AE in the inactivated vaccine group. This raises concerns about the risk-benefit ratio of mRNA-1010 in older adults, particularly given that the HAI titers for this age group against H3N2 and flu B are similar to those of the inactivated vaccine group. The higher proportion of systemic AEs and similar antibody responses suggest that the inactivated vaccine may be a better alternative for older adults. The authors should provide a discussion on the safety profile of mRNA-1010 in older adults and the potential trade-offs between the higher proportions of AEs and the similar or marginally higher immune responses of mRNA-1010 across different doses.

Author response and action taken: The manuscript has been revised to include further discussion of the potential advantages of mRNA-1010 among older adults in relation to any potential risks, such as increased reactogenicity events. Overall, we acknowledge that while mRNA-1010 had no safety concerns in this phase 1/2 trial, the vaccine was associated with increased local and systemic adverse reactions compared with Afluria. Additional assessments on the duration of protection and robustness of cellular immune responses, as well as evaluations on the efficacy and safety of mRNA-1010 in larger phase 3 trials, are aimed to provide more clarity on the overall benefit-risk profile of mRNA-1010 in this population.

Discussion (lines 371-380): *“Further, based on findings with mRNA-1273, an mRNA-based vaccine against SARS-CoV-2, mRNA vaccines may also induce strong cellular immune responses and prolonged germinal center reactions²⁷⁻³⁰ that can improve protection in older adults, a population at particular risk for infection and severe outcomes.¹⁰ Further laboratory assessments to evaluate these potential advantages are ongoing. While mRNA-1010 had an acceptable safety profile in this trial, transient solicited ARs were more common after mRNA-1010 than with the*

active comparator. Additional clinical trials are ongoing to further assess the safety, efficacy, and immunogenicity of this vaccine and a licensed influenza vaccine comparator (NCT05566639 and NCT05415462).

[...] Additional information on the safety and immunogenicity of mRNA-1010 will be gained from the final analyses of these 2 parts of the phase 1/2 trial, which will assess the longevity of immune responses through Day 181, as well as the phase 2 extension of this trial that will evaluate lower dose levels of mRNA-1010 in adults. Subsequent analyses will also include assessments of HA-specific T-cell and B-cell responses. Larger phase 3 trials of mRNA-1010 safety and immunogenicity in approximately 6000 adults aged ≥18 years (NCT05415462) as well as the safety and efficacy of mRNA-1010 in approximately 23000 adults aged ≥50 years (NCT05566639) are also currently underway."

Comment 3. Providing the study protocol and statistical analysis plan as supplementary materials would help to increase the transparency and rigor of the study, and would allow readers to better evaluate the validity and reliability of the results.

Author response and action taken: The study protocol and statistical analysis plan, redacted for proprietary information, have now been provided as supplementary files to accompany the manuscript.

Reviewer 3

Comment 1. This manuscript describes a first-in-human Phase 1/2 dose-finding trial of a quadrivalent influenza mRNA vaccine manufactured using a platform that has demonstrated good efficacy against the SARS-CoV-2 virus with an acceptable safety profile. The Phase 1 portion of the current study appears to have been conducted in Australia in 2021 and the Phase 2 in the US in 2021-2022. The dates of enrollment would be important to allow assessment of the role of recent exposure to wild-type virus, especially the B lineages.

Author response and action taken: We thank the Reviewer for their feedback and note that details on the study site locations (20 sites in the United States) are included in the Methods ('Trial Design and Participants' sub-section) while the dates of enrollment are included in the Results ('Participants' sub-section), in accordance with CONSORT guidelines.

Part 1 (lines 201-203): *"In Part 1 of the study, a total of 180 participants aged ≥ 18 years were randomly assigned to receive placebo or 1 dose of mRNA-1010 (50 μg , 100 μg , or 200 μg) between July 6, 2021, and August 18, 2021 (Figure 1)."*

Part 2 (lines 207-209): *"In Part 2 of the trial, a total of 501 participants aged ≥ 18 years were randomly assigned to receive a single dose of Afluria ($n=53$) or mRNA-1010 (25 μg , $n=152$; 50 μg , $n=149$; 100 μg , $n=147$) between November 10, 2021, and November 15, 2021; 498 participants received vaccination (Figure 1)."*

Comment 2. Unfortunately, the sample sizes of the two portions of the study are insufficient to support a clear distinction with respect to optimal dose (except as the higher doses are clearly associated with more frequent and severe reactogenicity, especially in younger adults). Furthermore, the use of an egg-based vaccine as the active comparator does not allow the assessment of relative immunogenicity between the mRNA platform and the more relevant comparator of a recombinant platform.

Author response and action taken: While participant sample sizes were insufficient to perform any statistical comparisons between groups in this phase 1/2 trial, we have now included 2 new supplemental figures that graphically display immunogenicity findings for all age groups

combined (adults ≥ 18 years of age). Supplement Figure 4 shows GMTs and GMFRs for each influenza strain for all age groups, while Supplement Figure 5 shows the ratio of GMTs for mRNA-1010 versus Afluria for all age groups combined. The increased participant sample size for these combined analyses allows for a clearer delineation of immunogenicity findings by mRNA-1010 dose level. We also note that phase 3 clinical trials with increased population sizes are currently underway (as described in the Discussion) and additional trials are planned to actively compare mRNA-1010 (50 μg) to non-egg-based influenza vaccines.

Comment 3. Of note, Figure 5 does not appear to present the immunogenicity induced by the comparator, despite the title of the figure - an apparent oversight.

Author response and action taken: We respectfully note that Figure 5 currently displays the ratio of GMTs induced by mRNA-1010 relative to Afluria at Day 29; we have amended the figure legend to further highlight this comparison to Afluria, as follows "Horizontal dotted line indicates a GMT ratio of 1, which reflects comparable GMTs between mRNA-1010 and Afluria."

Comment 4. In summary, the study certainly supports the conclusion that further evaluation is warranted. The magnitude and severity of reactogenicity may prove to be a challenge to annual repeated vaccination.

Author response and action taken: We thank the Reviewer for their feedback. Phase 3 trials on the safety, immunogenicity, and efficacy of mRNA-1010 in adults are currently underway and should provide further clarity on the risk-benefit of mRNA-1010 vaccination in this population.

Reviewers' Comments:

Reviewer #2:

Remarks to the Author:

Thank you for addressing the reviewer comments and providing the protocol and SAP. The revision and supplementary materials improve the paper's quality, and I have no further concerns.

Reviewer #3:

Remarks to the Author:

The authors have satisfactorily responded to the original reviewers' comments. My major remaining comment is that the use of a "standard dose" egg-based vaccine as the active control is somewhat less informative regarding the likely utility of this mRNA vaccine in the older adult population for whom better vaccines are especially needed. Comparison with the "high-dose" vaccine would be more relevant. This is particularly important since the reactogenicity is more problematic (if transient) than with existing vaccines. Additionally, if the issue of lack of egg-induced or cell-induced mutations is perceived to be an advantage for the mRNA platform, then the recombinant HA vaccine should be addressed.

Reviewer 2

Remarks to the Author: Thank you for addressing the reviewer comments and providing the protocol and SAP. The revision and supplementary materials improve the paper's quality, and I have no further concerns.

Author response and action taken: We thank the Reviewer again for their time and prior feedback on our manuscript, as well as their acknowledgment of having no further concerns.

Reviewer 3

Remarks to the Author: The authors have satisfactorily responded to the original reviewers' comments. My major remaining comment is that the use of a "standard dose" egg-based vaccine as the active control is somewhat less informative regarding the likely utility of this mRNA vaccine in the older adult population for whom better vaccines are especially needed. Comparison with the "high-dose" vaccine would be more relevant. This is particularly important since the reactogenicity is more problematic (if transient) than with existing vaccines. Additionally, if the issue of lack of egg-induced or cell-induced mutations is perceived to be an advantage for the mRNA platform, then the recombinant HA vaccine should be addressed.

Author response and action taken: We appreciate this feedback and note that additional clinical trials in adult populations are ongoing to further investigate the safety and immunogenicity of mRNA-1010 against other active comparators, including high-dose, recombinant protein, and adjuvanted influenza vaccines. The Discussion has been accordingly further revised to now state as such: *"Further studies will also evaluate mRNA-based seasonal influenza vaccines against enhanced influenza vaccines (ie, high-dose, recombinant protein, or adjuvanted vaccines) in adult populations (NCT05397223, NCT05606965, and NCT05333289)."*